# Effects of Pre- and Post-Carburizing Surface Modification on the Tribological and Adhesion Properties of Heat-Resistant KHR 45A Steel for Cracking Tubes

**DOI:** 10.3390/ma14133658

**Published:** 2021-06-30

**Authors:** Auezhan Amanov, Joo-Hyun Choi, Young-Sik Pyun

**Affiliations:** 1Department of Mechanical Engineering, Sun Moon University, Asan 31460, Korea; 2Department of Fusion Sciences and Technology, Sun Moon University, Asan 31460, Korea; pyoun@sunmoon.ac.kr; 3R&D Center, Boogong Industrial Co., Ltd., Asan 31435, Korea; joohyun@boogong.com

**Keywords:** heat-resistant KHR 45A steel, carburizing, tribology, adhesion, surface modification

## Abstract

In this study, the effects of ultrasonic nanocrystal surface modification (UNSM) technology on the tribological properties and scratch-induced adhesion behavior of a heat-resistant KHR 45A steel cracking tube, which is used for the pyrolysis process, were investigated. The main objective of this study is to investigate the effects of pre- and post-carburizing UNSM treatment on the tribological and adhesion performances of carburized domestic KHR 45A (A) steel and to compare the results with the existing carburized Kubota-made KHR 45A steel (B). A carburizing process was carried out on the polished and UNSM-treated KHR 45A steel substrates, which were cut out from the cracking tube, at 300 °C heat exposure for 300 h. The thickness of the carburizing layer was about 10 μm. UNSM technology was applied as pre- and post-carburizing surface treatment; both reduced the friction coefficient and wear rate compared to that of the carburized KHR 45A steel substrate. It was also found that the application of UNSM technology increased the critical load, which implies the improvement of adhesion behavior between the carburizing layer and the KHR steel substrate. The application of UNSM technology as pre- and post-carburizing surface treatment could help replace carburized Kubota-made KHR 45A steel (B) thanks to the improved tribological performance, enhanced scratch resistance, load bearing capacity, and adhesion of domestic KHR 45A (A) steel.

## 1. Introduction

The manufacturing process of heat-resistant steel tubes by the core centrifugal casting method is the most essential and common process in the petrochemical industry [1]. In this process, a molten metal is injected into the cylindrical mold rotating at high speed and then it is pushed to the inner surface of the mold by centrifugal force [2]. Heat-resistant steel tubes manufactured by general drawing and extrusion methods have room for shrinkage due to solidification from the outer and inner surfaces to the center of the product, whereas horizontal centrifugal casting has a solidification direction from the outside to the inside, and the growth of the metal structure generally occurs in the columnar direction. In addition, due to the characteristics of the centrifugal casting process, it is a very efficient process in producing cylindrical tubes due to the low scrap generation rate [3]. In fact, centrifugal casting processes are selected in fields that require high reliability, such as compressor cases, jet engines, and petrochemicals, due to the soundness of materials due to their dense metal structure.

Attention needs to be paid to the manufacturing process of ethylene because it is one of the main raw materials of the petrochemical industry, where a cracking tube plays a central role in the ethylene cracking furnace. Moreover, coking in the pyrolysis process has an influential effect on the dehydrogenation catalytic reaction of hydrocarbons by Fe or Ni due to the increase in heat transfer resistance and pressure loss on the inner surface of the cracking tube. It also tends to change the microstructure of the tube by carburization, resulting in volume expansion and heat [4]. In turn, this lowers the coefficient of expansion and causes material embrittlement leading to cracking tube breakage. Cracking tubes are exposed to high temperatures of up to approximately 1100 °C as they are used under such extreme conditions, and cracking coils are required to have high heat resistance and thermal efficiency [5]. In general, Ni-based heat-resistant steels are used to satisfy the high temperature strength and creep resistance of heat-resistant cracking tubes due to the strong catalytic action to form a filamentary coke (carbon deposition) on the inside surface of a cracking coil. The formation of coke shortens the service life of cracking tubes. These all increase the pressure drop in the cracking coil and lower the heat transfer efficiency, eventually causing the reduction in the operation efficiency of a thermal cracking furnace. In addition, when the reaction temperature is too high or exposure occurs for a long time, the decomposition reaction proceeds excessively, and amorphous coke is produced by the condensation polymerization reaction.

Kubota Co., Ltd. (Osaka, Japan) developed a high-performance Fe-based oxide dispersion reinforced alloy (Fe-20Cr-4.5Al-0.5Ti-0.5Y2O3) that significantly increased the operation efficiency of thermal cracking furnaces due to the greatly exceeded coking resistance and anti-carburization performance [6]. The continuous operation period of newly developed cracking tubes carburized at 1100 °C for 300 h was more than twice (20 years) that of the conventional material. The petrochemical industry demands a high formability such as heat-resistant mechanical properties, high-temperature creep properties, and sound thickness, but the current situation in Korea does not yet allow implementing such a shape by the centrifugal casting method. In addition, it is very important to secure durability and reliability for use in high temperature and harsh environments of petrochemical tube centrifugal casting parts, and there is no research and development related to materials and manufacturing process technology for this purpose. Therefore, a heat-resistant cracking steel tube with coking resistance and carburization resistance with a reliable new surface modification technology is in high demand because it alters the microstructure of a material surface through special treatment to add a new function to the base material. In the case of domestic heat-resistant cracking steel companies, in order to secure a high yield, they are using it under more severe conditions than the general recommended conditions of overseas component manufacturers. Due to this harsh atmosphere of domestic major companies, the domestic petrochemical industry has a short lifespan and is dependent on imports of all parts from Japan and Europe, resulting in a severe outflow of economic impact. Thus, it is urgent to develop new export routes based on the technology and price competitiveness. In addition, as investment in petrochemical facilities is expected to increase in Korea, the demand for centrifugal casting parts, which are the core of the facility, is expected to increase significantly.

As an alternative method, some other peening technologies such as shot peening (SP), surface mechanical attrition treatment (SMAT), and high-frequency mechanical impact (HFMI) can be used to increase the mechanical properties of various metallic materials [7,8,9]. As the trajectory of the ball is controlled by a computerized numerical control (CNC) machine during UNSM treatment, it involves higher kinetic energy than that in ultrasonic-base peening technologies, resulting in a thicker effective depth and deeper compressive residual stress with respect to depth from the top surface. In addition, one of the main advantages of UNSM over those ultrasonic-base peening technologies is the possibility of being applied to an inner surface of tubes and pipes. Accordingly, in this study, an ultrasonic nanocrystal surface modification (UNSM) technology was applied to the carburized cracking tubes, which are one of the core components of petrochemical facilities, to improve the tribological properties and adhesion between the carburized layer and the KHR 45A steel substrate. The main objective of this study is to investigate the effects of UNSM technology on the tribological and adhesion performances of carburized domestic KHR 45A steel and to compare the results with the carburized existing Kubota-made KHR 45A steel.

## 2. Materials and Methods

### 2.1. Material

In this study, carburized domestic (A) and Kubota-made (B) cracking tubes made of KHR 45A steel, which is one of the most popular cracking tube materials for high-temperature environments (>540 °C), were used. KHR 45A is a niobium-bearing high nickel–chromium alloy steel developed for ethylene pyrolysis service at temperature up to 1130 °C. The microstructure of KHR 45A steel consists of a ferritic matrix with precipitation of secondary phases (metal carbides and nitrides). Both KHR 45A (A) and (B) steels were fabricated using a centrifugal casting method and then were solution- and age-treated at 1518 °C. A carburizing process was carried out at a temperature of 650 °C, a pressure of 1 atm, and a gas flow rate of 100 cm^3^/min for an exposure time of 300 h on the square specimens, which were cut out from the cracking tube with an outer diameter of 129.4 mm and a thickness of 6.75 mm, with dimensions of 20 × 15 × 6 mm^3^. All the specimens were ultrasonically washed in water and rinsed in acetone, using an ultrasonic bath (SD-80H, Mujigae, Seoul, Korea) for 10 min. Subsequently, the plate specimens (nine pieces of each untreated and pre-UNSM-treated) with dimensions 20 × 15 × 5 mm^3^ were loaded onto platinum wires to be placed inside the furnace. The mechanical properties and chemical composition of cracking tubes made of KHR 45A steel are listed in Table 1 and Table 2, respectively. The main role of Al in the alloy is to form an oxide layer with increased resistance to oxidation at high temperatures, which completes the effect of chromium oxide formation [10]. Cross-sections of the sample were ground first and then polished using silicon carbide (SiC) sandpaper and a diamond suspension, and finally polished using a colloidal silica suspension. It was confirmed that the thickness of the carburizing layer was approximately 10 μm as shown in Figure 1.

### 2.2. UNSM Technology

UNSM is a surface severe plastic deformation (S^2^PD) technology, where a main source is a high-frequency ultrasonic impact. In general, UNSM technology generates a nanostructured layer in metallic materials due to the high contact pressure (up to 30 GPa) between the tungsten carbide (WC) ball and the workpiece. Thanks to the effect of high-frequency impact, severe surface hardening and high magnitude of compressive residual stress are involved at the top layer of the metallic materials. A schematic of UNSM technology is illustrated in Figure 2. It is important to mention here that a newly designed UNSM device can be applied to an inner surface of the cracking tube with a diameter of 131.8 mm. More details of UNSM technology along with a newly designed one for inner diameter treatment can be found in our previous studies [11,12,13]. In this study, UNSM technology was applied as a pre- and post-carburizing process under the treatment parameters listed in Table 3. It is important to note here that UNSM technology is usually applied to metallic materials, ceramics, coatings, and graphite, but there are no studies addressing the effects of UNSM technology on the mechanical properties, tribological performance, and scratch resistance of a carburizing layer with a thickness of about 10 μm. Figure 3 presents the cross-sectional microstructure and chemical composition mapping of the carburized KHR 45A steel subjected to UNSM treatment. It was found that the thickness of the carburizing layer decreased due to the effect of UNSM treatment, and the absence of complex compounds, oxide crusts, etc., at the interface was confirmed.

### 2.3. Tribological and Scratch Tests

Tribological properties were evaluated in the ball-on-disc configuration in accordance with the ASTM G133-05 standard using a set-up (Anton Paar GmbH, Graz, Austria) under the conditions listed in Table 4. An SAE 52100 bearing steel counterface ball with a diameter of 12.7 mm was selected due to its higher hardness than carburized KHR 45A steel. Progressive scratch tests were carried out by a scratch tester equipped with a Rockwell diamond probe with a tip radius of 200.0 µm at a progressive load in the range of 30.0 to 150.0 N with a scratch speed of 20.0 mm/min over a total scratch length of 12.2 mm. At least 2–3 wear track and scratch grooves were produced, and the obtained results were found to be very reproducible with acceptable standard deviation.

### 2.4. Characterization

Surface roughness and surface hardness were measured five times using a two-dimensional (2D) surface profilometer (SJ-200, Mitutoyo, Kawasaki, Japan) and a micro-Vickers hardness tester (Mitutoyo MVK-E3, Japan) at a load of 300 gf with a dwell time of 12 s. The surface morphology, wear track, and scratch grooves were characterized by a scanning electron microscope (SEM; JEOL 6061, Tokyo, Japan) equipped with an energy dispersive X-ray spectroscope (EDX: JED2300, Tokyo, Japan) at a voltage of 15 kV. Bruker D8 Advance XRD (X-ray diffractometer, Karlsruhe, Germany) was employed using CuKα radiation (1.5406 Å) in the range of 30° to 80° 2θ angle at a step of 0.01°/s.

## 3. Results and Discussion

### 3.1. Surface Roughness and Hardness

Figure 4a shows the comparison of surface roughness *(R_a_—arithmetical mean deviation and R_z_—maximum peak to valley height of the profile)* values of the carburized KHR 45A steel. It was found that both the *R_a_* and *R_z_* values of the carburized KHR 45A (A) steel were reduced from 2.7 µm to 0.2 µm and from 12.5 µm to 1.3 µm, respectively, after UNSM treatment. Pre-carburizing UNSM treatment increased the surface roughness *R_a_* and *R_z_* values of the carburized KHR 45A (A) steel from 2.7 µm to 3.2 µm and 12.5 µm to 15.1 µm, while a post-carburizing UNSM treatment reduced the surface roughness *R_a_* and *R_z_* values of the carburized specimen from 3.2 µm to 0.3 µm and 15.1 µm to 1.4 µm, respectively. The surface roughness *R_a_* and *R_z_* values of the carburized KHR 45A (B) steel were also reduced from 2.4 µm to 0.1 µm and 13.1 µm to 0.9 µm, respectively, after UNSM treatment. As a result, the surface roughness *R_a_* and *R_z_* values of the carburized KHR 45A (A) and KHR 45A (A) + UNSM steels were found to be higher than those of the KHR 45A (B) and KHR 45A (B) + UNSM steels. Pre- and post-carburizing UNSM treatment led to slightly higher surface roughness compared to the surface roughness of those post-carburized UNSM-treated KHR 45A (A) and KHR 45A (B) steels. It is worth mentioning here that the surface roughness of the carburized layer with a thickness of about 10 µm can be easily controlled by UNSM treatment to satisfy customers’ needs. As a consequence, it was found that the pre- and post-carburized UNSM treatment can reduce the surface roughness by an order of magnitude. During UNSM, the carburized layer deformation took place by the combined action of sliding and striking of the WC ball, where the redundant materials under the profile peaks were pushed to fill in the valleys. Thus, a very smooth surface can be obtained that is effective in avoiding stress concentration and retarding the initiation of tensile cracks during scratching [14].

Figure 4b shows the comparison of surface hardness of the carburized KHR 45A steel. It is obvious that the surface hardness of the carburized KHR 45A (A) steel was increased from 286 HV to 322 HV after UNSM treatment, which resulted in a corresponding 12% increase. Prior to carburizing, the polished KHR 45A (A) steel was first treated by UNSM technology and subsequently carburized and then again was treated by UNSM technology. Pre-carburizing UNSM treatment increased the surface hardness of the carburized KHR 45A (A) steel from 286 HV to 355 HV, while a post-carburizing UNSM treatment further increased the surface hardness of the carburized KHR 45A (A) steel from 355 HV to 394 HV, which resulted in a corresponding 10% increase. The surface hardness of the carburized KHR 45A (B) steel was also increased from 178 HV to 314 HV after UNSM treatment. As a result, the surface hardness of the carburized KHR 45A (A) and KHR 45A (A) + UNSM steels was found to be higher than that of the KHR 45A (B) and KHR 45A (B) + UNSM steels. Pre- and post-carburizing UNSM treatment demonstrated higher surface hardness than the surface hardness of those post-carburizing UNSM-treated KHR 45A (A) and KHR 45A (B) steels. An increase in surface hardness of the carburized specimens subjected to pre- and post-UNSM treatment may be attributed to the deformation hardening and refined grain size of the KHR 45A steel and carburized layer. This hardening mechanism can be explained by Hall–Petch expression, where a grain size is the most important factor [15]. The effective depth in terms of hardness was found to be about 5–6 µm, where an increase in hardness along the depth gradually decreased because of the gradient distribution of plastic strain with respect to depth. In addition, the UNSM technology increases the dislocation density significantly in the plastically deformed layer, resulting in high strength and hardness due to the grain size refinement [16,17]. The grain size refinement mechanism may be attributed to the formation of new dislocations, where the accumulation of dislocations at low- and high-angle grain boundaries leads to the variation of the orientation associated with a part of a grain, resulting in grain size refinement.

### 3.2. XRD Pattern

Figure 5 shows the comparison of XRD patterns obtained from the surface of the carburized and post-carburized UNSM-treated KHR 45A steels. It was found that the intensity of diffraction peak of the KHR 45A (A) steel was reduced with no new phase identification after UNSM treatment. In addition, the full width at half-maximum (FWHM) value of the diffraction peak (111) widened from 0.21 ± 0.00116 to 0.39 ± 0.00363 after UNSM treatment. A reduction in intensity is associated with the change in grain size, which confirms the refinement of coarse grains into nano-sized grains after UNSM treatment, while a widening in FWHM may be attributed to the increase in lattice strain [18]. In addition, a diffraction peak position shift after UNSM treatment was noticed to higher angles due to the changes in tensile residual stress into compressive residual stress [19]. Furthermore, as shown in Figure 5, the presence of two different types of chromium carbides (Cr_23_C_6_ and Cr_7_C_3_) and cementite (Fe_3_C) was identified on the surface of the carburized and UNSM-treated KHR 45A steel. It is believed that the cementite was formed due to the transformation of Fe from the steel substrate made of KHR 45A.

### 3.3. Tribological Performance

Figure 6a shows the comparison of the average friction coefficient along with the standard deviation of the carburized KHR 45A steels. The friction coefficient of the carburized KHR 45A (A) steel was reduced from 0.232 to 0.191 after UNSM treatment, which resulted in a corresponding 17.7% reduction. Pre-carburizing UNSM treatment increased the friction coefficient of the carburized KHR 45A (A) steel from 0.232 to 0.264, while post-carburizing UNSM treatment reduced the friction coefficient of the carburized KHR 45A (A) steel from 0.264 to 0.251, which resulted in a corresponding 5% reduction. The friction coefficient of the carburized KHR 45A (B) steel was reduced from 0.295 to 0.142 after UNSM treatment. As a result, the friction coefficient of the carburized KHR 45A (A) steel was found to be lower than that of the carburized KHR 45A (B) steel, while the carburized KHR 45A (B) + UNSM steel exhibited a lower friction coefficient than that of the carburized KHR 45A (A) + UNSM steel. Pre- and post-carburizing UNSM treatment demonstrated slightly higher and lower friction coefficients than those of the post-carburizing UNSM-treated KHR 45A (A) and KHR 45A (B) steels, respectively. The KHR 45A (B) and KHR 45A (B) + UNSM steels demonstrated smaller standard deviations than those of the pre- and post-carburized KHR 45A (A) and KHR 45A (A) + UNSM steels, implying a stable friction coefficient with low fluctuation throughout the testing period. As shown in Figure 6a, as the surface roughness is an essential property of friction surfaces; the UNSM treatment played an influential role in reducing the friction coefficient especially of the carburized KHR 45A (B) + UNSM steel, where a reduction in surface roughness after UNSM treatment reduced the contact stress due to the increased number of asperities, and thus the sliding friction reduced [11,20].

A comparison of the wear rate of the carburized KHR 45A steels calculated taking into account the wear track width and depth, and applied load and total sliding distance is shown in Figure 6b. It was found that the wear rate of the carburized KHR 45A (A) steel was reduced from 0.00955 mm^3^/N × mm and 0.00674 mm^3^/N × mm after UNSM treatment. As can be seen from Figure 6b, pre-carburizing UNSM treatment reduced the wear rate of the carburized KHR 45A (A) steel from 0.00585 mm^3^/N × mm to 0.00465 mm^3^/N × mm, while post-carburizing UNSM treatment further reduced the wear rate of the carburized KHR 45A (B) steel from 0.0128 mm^3^/N × mm to 0.0103 mm^3^/N × mm, which resulted in a corresponding 19.6% reduction. As a result, the wear rate of the carburized KHR 45A (A) and KHR 45A (A) + UNSM steels was found to be lower than that of the KHR 45A (B) and KHR 45A (B) + UNSM steels. Pre- and post-carburizing UNSM treatment demonstrated better wear resistance than the wear resistance of those the post-carburized UNSM-treated KHR 45A (A) and KHR 45A (B) steels. An increase in wear resistance of the carburized KHR 45A steel subjected to pre- and post-UNSM treatment may be attributed to the increase in hardness, which is associated with grain size refinement of KHR 45A steel and the carburized layer as explained earlier in Section 3.1. In general, many metallic materials rely on grain size refinement to ensure high hardness, which has a strong correlation with wear resistance, i.e., high hardness of the material provides a high wear resistance [21,22]. Lindroos et al. reported that the increased surface hardness will have a beneficial effect on wear resistance only if the material retains its toughness in the deformed layer [23]. Moreover, not only increases in surface hardness, but also changes in fracture toughness and work-hardening behavior after UNSM treatment have a significant role in increasing the wear resistance of metallic materials. Hence, comprehensive mechanical testing that provides some information about the changes in toughness and work-hardening behavior after UNSM treatment needs to be performed to ensure the influence of the increase in surface hardness on wear resistance enhancement.

### 3.4. Scratch-Induced Adhesion Performance

It is a challenge to improve the adhesion properties of carburized protective layers with a thickness of about 10 µm by pre-carburizing UNSM treatment of the steel substrate. A combination of UNSM technology and the carburizing process, which can be considered a duplex treatment is of interest to investigate the effect on the adhesion behavior between the carburizing layer and the steel substrate. Figure 7 shows the comparison of scratch-induced friction force and acoustic emission of the carburized KHR 45A steels. The solid black line in Figure 7 represents the incremental load applied during scratching. As shown in Figure 7a,a1, the friction force of the carburized KHR 45A (A) steel was found to be lower than that of the post-carburized UNSM-treated KHR45 A (A) steel at the beginning of scratching. Acoustic emission of the carburized KHR 45A (A) steel showed an abrupt increase up to 15% at the beginning of scratching and then a gradual decrease with increasing scratching distance (see Figure 7a). This implies that the carburized layer was fractured very fast at an applied load of 36 N, which was a critical load for the carburized KHR 45A (A) steel. As shown in Figure 7a1, acoustic emission of the post-carburized UNSM-treated KHR 45A (A) steel was increased step-by-step up to 9% at an applied of 48 N, which was a critical load for the post-carburized UNSM-treated KHR 45A (A) steel, and then it gradually reduced with increasing scratching distance. Based on the scratching data, the critical load of the carburized layer was increased from 36 N to 48 N after UNSM treatment, which resulted in a corresponding 25% increase. Scratch-induced friction force and acoustic emission of the pre-carburizing, and pre-, and post-carburized UNSM-treated KHR 45A (A) steels are shown in Figure 7b,b1. It was found that the pre-carburizing UNSM treatment increased the friction force of the carburized KHR 45A (A) steel (see Figure 7b), while both pre- and post-carburizing UNSM treatment reduced the friction force of the carburized KHR 45A (A) steel compared to the pre-carburizing UNSM treatment (see Figure 7b1). An increase in the friction force of the pre-carburizing UNSM treatment may be attributed to the adhesion between the carburized layer and the substrate, and plowing occurred thanks to the diamond indenter. Fortunately, both the pre- and pre-and post-carburizing UNSM treatment reduced the acoustic emission significantly compared to the carburized and post-carburized UNSM-treated KHR 45A (A) steels. It was found that the acoustic emission of the pre-carburized KHR 45A (A) steel was negligible and kept increasing gradually at a scratching distance of 5.1 mm and then reached a maximum failure point at a scratching distance of 5.8 mm, which corresponds to the critical load of 84 N. Then, it gradually decreased with increasing scratching distance (see Figure 7b). This suggests that the pre-carburizing UNSM treatment had an influential effect on the adhesion behavior of the KHR 45A (A) steel and carburized layer. Figure 7b1 shows the acoustic emission of the pre- and post-carburized UNSM-treated KHR 45A (A) steels. Obviously, a negligible low acoustic emission of the pre- and post-carburized KHR 45A (A) steels was observed at the beginning of scratching. It confirms that the pre- and post-carburized KHR 45A (A) steel had better adhesion behavior with no failure compared to that of the pre-carburized and post-carburized KHR 45A (A) steels. The friction force and acoustic emission of the carburized KHR 45A (B) steel are shown in Figure 7c,c1. It was found that the friction force of the carburized KHR 45A (B) steel was found to be lower than that of the UNSM-treated KHR 45A (B) steel in the second half of scratching. Acoustic emission of the carburized KHR 45A (B) steel showed an abrupt increase up to 20% at the beginning of scratching and then a gradual decrease with increasing scratching distance (see Figure 7c). It was confirmed that the carburized layer was fractured once the diamond intender came into contact with the carburized layer, at an applied load of 30 N, which was a critical load for the carburized KHR 45A (B) steel. As shown in Figure 7c1, acoustic emission of the post-carburized UNSM-treated KHR 45A (B) steel was increased step-by-step up to 19% at an applied of 56 N, which was a critical load for the post-carburized UNSM-treated KHR 45A (B) steel, and then gradually reduced step-by-step with increasing scratching distance. Hence, it was confirmed that the critical load of the carburized KHR 45A (B) steel was increased from 30 N to 56 N after UNSM treatment, which resulted in an almost twofold increase. As a result, the carburized KHR 45A (B) steel showed the lowest friction force, while the pre-carburized KHR 45A (A) steel showed the highest friction force among the KHR 45A steels. Interestingly, the results of the friction force and acoustic emission of the specimens were controversial, where the carburized KHR 45A (B) steel with the lowest friction force showed the fastest failure with a critical load of 30 N, while the pre-carburized KHR 45A (A) steel had much better acoustic emission except for the pre- and post-carburized KHR 45 (A) steel. In summary, the carburized KHR 45A (A) steel had a better adhesion property including the critical load compared to the KHR 45A (B) steel. Furthermore, the application of UNSM treatment pre- and post-carburizing further improved the adhesion property including critical load with no failure for the pre- and post-carburized KHR 45A (A) steel. Actually, the surface roughness of the KHR 45A steel substrate affects the adhesion that may lead to a blistering, flaking and delamination [24]. Menezes et al. investigated the effect of shot peening that was applied to the substrate as a pre-treatment that led to a greater hardening depth with low surface roughness [25]. Furthermore, it was reported earlier that despite the substrate being treated by plasma nitriding as a pre-treatment prior to the deposition of a coating by physical-vapor deposition (PVD), coating failure occurred at the interface between the coating and the substrate [26]. Based on the obtained results, it can be assured that the UNSM technology increased the critical load of the carburizing layer that signifies the improvement in the adhesion of the carburized layer to the KHR 45A steel substrate.

SEM images of the ends of scratch grooves showing scratch-induced grooves formed on the surface of the carburized KHR 45A steels are shown in Figure 8. A remarkable influence of UNSM technology on the scratch resistance of the carburizing layers was observed. Specifically, it was observed that the width of the scratch groove formed on the surface of the carburizing layer of all KHR 45A (A) and (B) steels was found to be wider than that of the post-carburized UNSM-treated KHR 45A (A) and (B) steels at the end of scratching, where the latter showed relatively smooth scratch grooves and edges. The average scratch groove width and residual penetration depth obtained from the ends of scratch grooves is presented in the inset of Figure 8. It is certain according to the scratch groove depth profile (not shown here) that the indenter reached into the substrate. Furthermore, the scratch resistance at the end of scratching for the carburized KHR 45A steels was calculated using Equation (1) in accordance with the ASTM G171-03 standard [27]:(1)HSp=8Nπw2
where *HS_p_*—the scratch hardness number (GPa), *N*—the applied normal load (N), and *w*—the scratch width (m). The maximum applied load of 150 N at the end of scratching was used as an *N* to calculate the scratch resistance as the incremental *N* was applied during scratching. It is obvious from the equation above that the narrower the scratch width, the higher the scratch hardness number, which leads to a high scratch resistance. The calculated scratch hardness number results of the carburized KHR 45A steels are also presented in the inset of Figure 8. In addition, the results of plastic resistance (PR) at the end of scratching for the carburized KHR 45A steels can be seen in the inset of Figure 8. It was shown that the scratch resistance had a strong correlation with scratch hardness, where the highest surface hardness showed shallower residual penetration depth (scratch hardness is proportional to the maximum progressive load and residual width of the scratch). The carburized KHR 45A (A) steel had the worst scratch resistance with the deepest residual penetration depth, while the carburized KHR 45A (B) steel had the fastest failure according to the AE signal as shown in Figure 7.

It is also observable from Figure 8a1–c1 that the post-carburized UNSM-treated KHR 45A (A) and (B) steels exhibited smoother scratch grooves compared to those of the carburized KHR 45A (A) and (B) steels (see Figure 8a–c). As shown in Figure 8a, the formation of some tensile cracks was noticed inside the scratch groove of the carburized KHR 45A (A) and (B) steels, where fewer tensile cracks were observed only inside the scratch groove of the post-carburized UNSM-treated KHR 45A (B) steel as shown in Figure 8c1. These tensile cracks on the surface of the carburized as well as post-carburized UNSM-treated KHR45A (B) steel substrates were formed because of the introduced tensile stress at the back side of the indenter during scratching. Moreover, it is clear that the pile-up of the scratch grooves that formed on the surface of the post-carburized UNSM-treated KHR 45A steel substrates was more pronounced than that of the carburized KHR 45A (B) steel substrates due to the introduced high and deep compressive residual stress. Based on a comparison of the scratch grooves of the KHR 45 steel substrates, it was obvious that the post-carburized UNSM-treated KHR 45A (B) steel substrate had no or fewer tensile cracks. Hence, the post-carburized UNSM-treated KHR 45A (B) steel substrate had better scratch resistance at a progressive load in comparison with the carburized KHR 45A (B) steel substrate. Pre- and post-carburizing UNSM treatments led to a noticeable improvement in the scratch resistance of the carburized layer, where pre-UNSM treatment resulted in the best scratch resistance with narrower scratch grooves and fewer pile-ups due to the increased hardness of KHR 45A (A) steel substrate and the carburizing layer. Scratching was performed at a progressive load, resulting in cohesive failure, which is the result of an increase in elastic and plastic deformation until cracks are initiated in the carburizing layer. Subsequently, an adhesive failure, which is the delamination of the carburizing layer from the KHR 45A steel substrate, took place. Failure mechanisms of carburized layers strongly depend on important factors such as surface roughness, hardness of both carburized layer and KHR 45A steel, scratching conditions, and the carburized layer thickness, etc. [28].

## 4. Conclusions

The industrial requirements for satisfactory mechanical, tribological, and scratch performances of a carburized KHR 45A steel are sufficient adhesion at the contact interface between the carburizing layer and the substrate and the ability of the substrate to support the carburized layer. Having controlled the surface and hardness of the KHR 45A steel substrate by UNSM technology that was applied as a pre-treatment, carburizing improved the load carrying capacity of the steel substrate that in turn was able to effectively improve the adhesion behavior at the contact interface. Both the *R_a_* and *R_z_* values of the carburized KHR 45A (A) steel were reduced by more and less than 1000%, respectively. Pre-carburizing UNSM treatment increased the surface roughness *R_a_* and *R_z_* values of the carburized KHR 45A (A) steel by about 118.5% and 120.8%, while a post-carburizing UNSM treatment reduced the surface roughness *R_a_* and *R_z_* values of the carburized specimen by more than 1000%. The surface roughness *R_a_* and *R_z_* values of the carburized KHR 45A (B) steel were also reduced by about more than 2400% and 1450%, respectively, after UNSM treatment. The surface hardness of the carburized KHR 45A (A) steel was increased by 12%. Pre-carburizing UNSM treatment increased the surface hardness of the carburized KHR 45A (A) by 19.5%, while a post-carburizing UNSM treatment further increased the surface hardness of the carburized KHR 45A (A) steel by 10%. The surface hardness of the carburized KHR 45A (B) steel was also increased by 43.4% after UNSM treatment. The friction coefficient of the carburized KHR 45A (A) steel was reduced by 17.7%. Pre-carburizing UNSM treatment increased the friction coefficient of the carburized KHR 45A (A) steel by 13.7%, while post-carburizing UNSM treatment reduced the friction coefficient of the carburized KHR 45A (A) steel by 5%. The friction coefficient of the carburized KHR 45A (B) steel was reduced more than two times under UNSM treatment. The wear rate of the carburized KHR 45A (A) steel was reduced by 29.5% after UNSM treatment. A pre-carburizing UNSM treatment reduced the wear rate of the carburized KHR 45A (A) steel by 20.6%, while a post-carburizing UNSM treatment further reduced the wear rate of the carburized KHR 45A (B) steel by 19.6%. Furthermore, UNSM technology improved the adhesion behavior of the carburized layer–substrate by the inter-diffusion between the iron and carbon at the interface. As a result, the application of UNSM technology to a carburized heat-resistant KHR 45A steel cracking tube led to an improvement in tribological and adhesion properties, which may be attributed to the reduction in surface roughness and increase in surface hardness of both the carburized layer and KHR 45A steel cracking tube. Both pre- and post-carburizing UNSM treatment had a beneficial effect on the wear enhancement and scratch resistance due to the presence of a hardened surface layer and also the difference in surface roughness of the surface of KHR 45A steel cracking tube.

## Figures and Tables

**Figure 1 materials-14-03658-f001:**
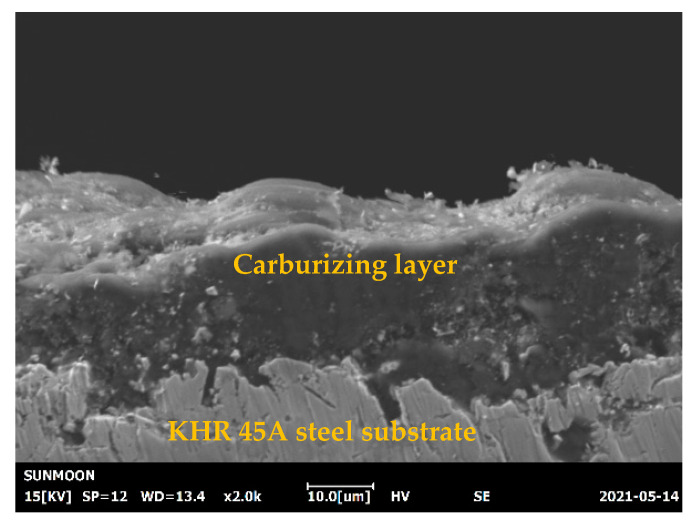
Cross-sectional SEM image showing the thickness of the carburizing layer on heat-resistant KHR 45A steel substrate.

**Figure 2 materials-14-03658-f002:**
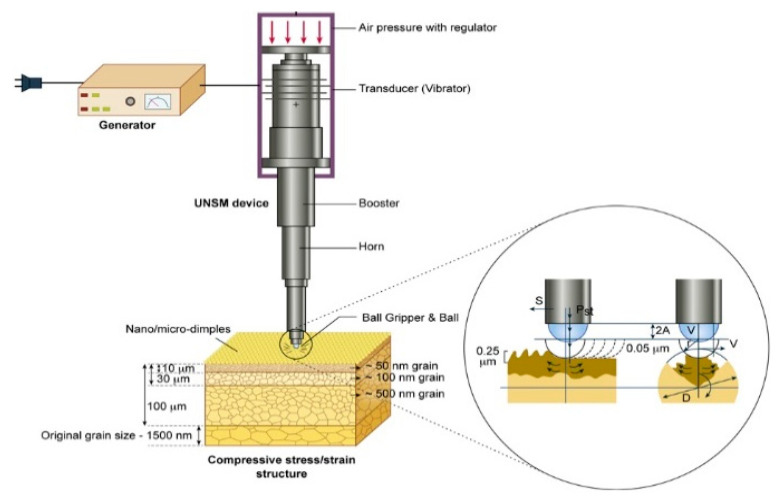
Schematic view of UNSM technology.

**Figure 3 materials-14-03658-f003:**
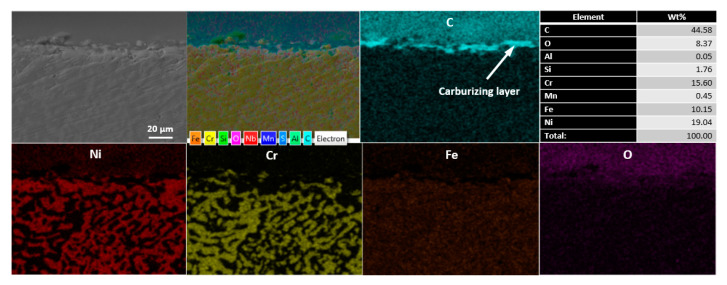
Cross-sectional SEM image and chemical composition mapping of the carburized KHR 45A steel subjected to UNSM treatment.

**Figure 4 materials-14-03658-f004:**
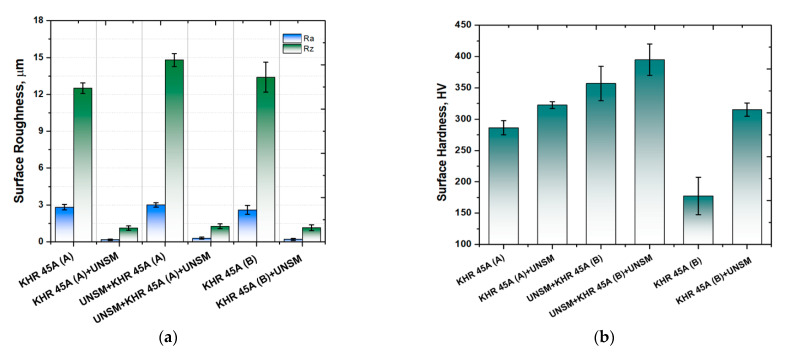
Comparison of surface roughness (**a**) and surface hardness (**b**) of the carburized, pre-, and post-carburized UNSM-treated KHR 45A steels.

**Figure 5 materials-14-03658-f005:**
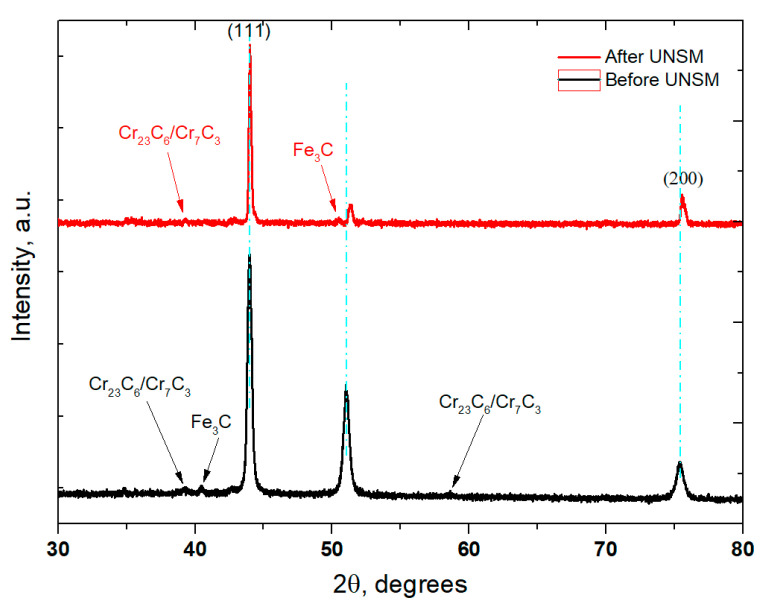
XRD patterns obtained from the surface of the carburized and post-carburized UNSM-treated KHR 45A steel (A).

**Figure 6 materials-14-03658-f006:**
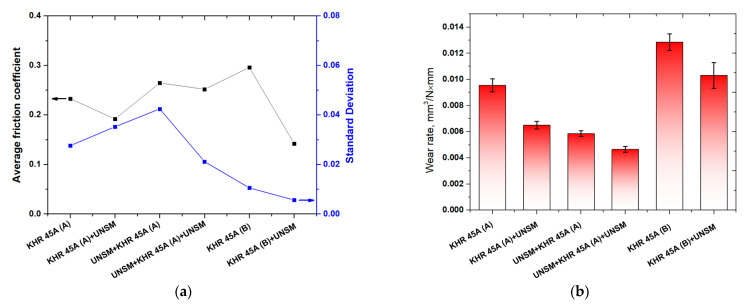
Comparison of the average friction coefficient along with the standard deviation (**a**) and wear rate (**b**) of the carburized, pre-, and post-carburized UNSM-treated KHR 45A steels.

**Figure 7 materials-14-03658-f007:**
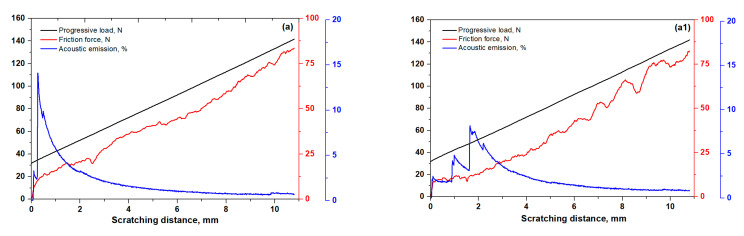
Comparison of scratch-induced friction force and acoustic emission of the carburized, pre-, and post-carburized UNSM-treated KHR 45A steels. (**a**) carburized KHR 45A (A), (**b**) pre-carburizing UNSM-treated KHR 45A (A), (**c**) carburized KHR 45A (B), (**a1**) carburized KHR 45A (A) + UNSM, (**b1**) UNSM + carburized KHR 45A (A) + UNSM and (**c1**) carburized KHR45A (B) + UNSM.

**Figure 8 materials-14-03658-f008:**
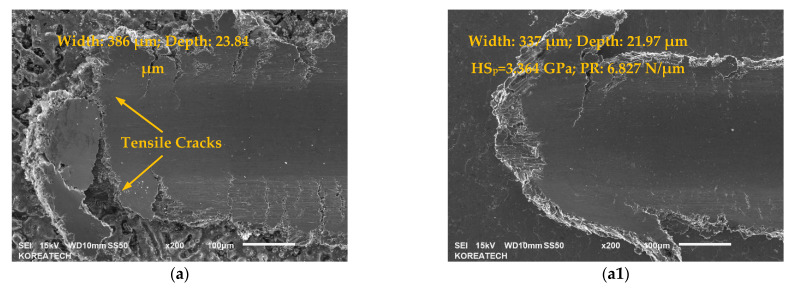
SEM images of the ends of scratch grooves showing scratch-induced grooves formed on the surface of the carburized, pre-, and post-carburized UNSM-treated KHR 45A steels. (**a**) carburized KHR 45A (A), (**b**) pre-carburizing UNSM-treated KHR 45A (A), (**c**) carburized KHR 45A (B), (**a1**) carburized KHR 45A (A) + UNSM, (**b1**) UNSM+ carburized KHR 45A (A) + UNSM and (**c1**) carburized KHR45A (B) + UNSM.

**Table 1 materials-14-03658-t001:** Mechanical properties of KHR 45A steel obtained at 21 °C.

**Tensile Strength, MPa**	**Yield Strength, MPa**	**Elastic Modulus, GPa**	**Impact, KV/Ku (J)**	**Poisson’s Ratio**	**Elongation, %**
517	282	171.7	23	0.294	41

**Table 2 materials-14-03658-t002:** Chemical composition of KHR 45A steel in wt.%.

**Fe**	**Cr**	**C**	**Ti**	**Mn**	**Si**	**Ni**	**S**	**P**	**Mo**	**Nb**	**Al**
17.62	18.84	0.40	0.95	0.02	0.06	53.64	0.002	0.003	3.08	1.8	0.53

**Table 3 materials-14-03658-t003:** UNSM treatment parameters.

**Frequency, kHz**	**Amplitude, µm**	**Speed, mm/min**	**Load, N**	**Feed-Rate, µm**	**Ball Diameter, mm**	**Ball Material**
20	50	2000	50	70	2.38	WC

**Table 4 materials-14-03658-t004:** UNSM treatment parameters.

**Applied Load, N**	**Stroke, mm**	**Linear Speed, cm/s**	**Frequency, Hz**	**Sliding Distance, m**
10	4	10	4	100

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
