# Peer review of "Effects of Pre- and Post-Carburizing Surface Modification on the Tribological and Adhesion Properties of Heat-Resistant KHR 45A Steel for Cracking Tubes"

_materials, 2021, doi:10.3390/ma14133658_

Round 1
Reviewer 1 Report
The topic is well developed, the measurements were well organized. It is necessary to add:
- in the introduction to add an overview of other methods, e.g. HFMI method, or describe the difference compared to UNSM
- what specimens were used in these analyzes? (dimensions, shape, number)
- in conclusions: add numerical expression of improved properties after UNSM application in%
- how the UNSM would be applied to the inner surface of the tube, or from what diameter it could be used

Author Response
Our point-by-point responses to the reviewer`s comments are attached.

Reviewer 2 Report
Title.
Effects of Pre- and Post-Carburizing Surface Modification on the Tribological and Adhesion Properties of Heat Resistant KHR 45A Steel for Cracking Tube
Overview:
The paper presents the effects on surface hardness and roughness in the case of KHR 45A alloy following surface treatments by carburizing and ultrasonic nanocrystal surface modification (UNSM) processing.
The authors showed that the hardness of the treated and processed surfaces increased by 12 and 10% respectively after the treatments performed. They claim that the best results were obtained in the case of ultrasonic processing before and after carburizing.
A series of explanations are proposed to demonstrate the observed hardening effects, based mainly on reports from the literature.
The paper is interesting and can bring elements of progress in terms of the combined treatment applied on the surfaces of special purpose alloys, that working in severe working conditions.
General recommendation:
Analyzes of chemical composition and microstructure performed in cross section on the treated layer are absolutely necessary to determine the effects of hardening (both by the formation of complex compounds, oxide crusts or multiplication of dislocations).
In the literature, this Fe-based alloy, which contains 20% wtCr and 4.5% wtAl is ferritic, not austenitic as specified in the paper. The Al content of this alloy has the role of forming an oxide layer with increased resistance to oxidation at high temperatures, which completes the effect of chromium oxide formation.
See: https://www.researchgate.net/publication/304562671_Hafnium_influence_on_the_microstructure_of_FeCrAl_alloys).
Please make the necessary corrections and add microstructure (SEM) and chemical composition analyzes along cross section of the treated interface to demonstrate and complete your observations and conclusions from the paper.

Author Response

(The authors gave the same response as above.)

Round 2
Reviewer 1 Report
The article has been suitably modified and supplemented. In this form it is suitable for publication.
Reviewer 2 Report
Accept in present form